# Solvent-Free Desulfurization System to Produce Low-Sulfur Diesel Using Hybrid Monovacant Keggin-Type Catalyst

**DOI:** 10.3390/molecules25214961

**Published:** 2020-10-27

**Authors:** Fátima Mirante, Baltazar de Castro, Carlos M. Granadeiro, Salete S. Balula

**Affiliations:** LAQV-REQUIMTE, Departamento de Química e Bioquímica, Faculdade de Ciências, Universidade do Porto, 4169-007 Porto, Portugal; fatima.mirante@fc.up.pt (F.M.); bcastro@fc.up.pt (B.d.C.); cgranadeiro@fc.up.pt (C.M.G.)

**Keywords:** polyoxometalate, pseudo-monophasic desulfurization, fuel, sulfur oxidation, hydrogen peroxide

## Abstract

Two quaternary ammonium catalysts based on the monovacant polyoxotungstate ([PW_11_O_39_]^7−^, abbreviated as PW_11_) were prepared and characterized. The desulfurization performances of the PW_11_-based hybrids (of tetrabutylammonium and trimethyloctadecylammonium, abbreviated as TBA[PW_11_] and ODA[PW_11_], respectively), the corresponding potassium salt (K_7_PW_11_O_39_, abbreviated as KPW_11_) and the peroxo-compound (TBA-PO_4_[WO(O_2_)_2_], abbreviated as TBA[PW_4_]) were compared as catalysts for the oxidative desulfurization of a multicomponent model diesel (2000 ppm S). The oxidative desulfurization studies (ODS) were performed using solvent-free systems and aqueous H_2_O_2_ as oxidant. The nature of the cation in the PW_11_ catalyst showed to have an important influence on the catalytic performance. In fact, the PW_11_-hybrid catalysts showed higher catalytic efficiency than the peroxo-compound TBA[PW_4_], known as Venturello compound. TBA[PW_11_] revealed a remarkable desulfurization performance with 96.5% of sulfur compounds removed in the first 130 min. The reusability and stability of the catalyst were also investigated for ten consecutive ODS cycles without loss of activity. A treated clean diesel could be recovered without sulfur compounds by performing a final liquid/liquid extraction diesel/EtOH:H_2_O mixture (1:1) after the catalytic oxidative step.

## 1. Introduction

The deep desulfurization of transportation fuels has received worldwide attention as a result of the increasingly rigid regulations concerning sulfur content in fuels. The traditional hydrodesulfurization (HDS) technology requires severe operational conditions and is less efficient in removing the aromatic sulfur compounds (dibenzothiophene and derivatives) present in fuels [1,2]. Different complementary desulfurization processes have been studied based on extraction, adsorption, oxidation or even biological methods [3,4,5]. The combination of extraction and oxidation steps, in the so-called extractive and catalytic oxidative desulfurization (ECODS) method, allows a highly efficient removal of refractory sulfur compounds under mild and eco-sustainable conditions [6,7,8,9,10,11]. Recently, there has been an increasing effort in developing solvent-free desulfurization systems [3]. These systems are intended to still efficiently remove organosulfur compounds from fuels with the environmental and economic advantages of avoiding the use of the traditional volatile organic solvents [12,13]. Jia and co-workers have proposed a solvent-free ODS system using a MoO_3_/γ-Al_2_O_3_ catalyst and H_2_O_2_ as oxidant [13]. Complete oxidation of dibenzothiophene (DBT) and 4,6-dimethyldibenzothiophene (4,6-DMDBT) could be reached after just 15 min of reaction. Regeneration of the catalyst was achieved by washing with methanol for reuse in consecutive cycles, although some loss of activity could be observed.

Polyoxometalates (POMs) are versatile compounds with peculiar physicochemical properties that enable their application in diverse fields such as magnetism, luminescence, medicine and catalysis [14]. Over the years, POMs and Keggin-type structures, in particular, have become increasingly used in acid and oxidative catalysis due to their remarkable activity and stability [15]. Our research group has been investigating novel POM-based eco-sustainable desulfurization systems for the production of sulfur-free fuels [16,17,18,19,20,21,22]. Recently, we have also been developing solvent-free systems able to match the desulfurization performance of biphasic systems while allowing the use of more sustainable extraction solvents for the final removal of oxidized products (e.g., water) [23,24,25]. The preparation of organic-POM hybrids by introduction of different organic cations (ionic liquids, surfactants and polymers) has proved to be a simple methodology for the heterogenization of active homogenous POMs while maintaining their catalytic performance [26,27,28]. Some examples can be found in the literature dealing with the application of these organic-POM hybrids in oxidative desulfurization of model and real fuels [29,30,31,32,33,34,35,36,37,38,39]. Campos-Martin et al. have recently reported an hybrid POM composed by a vanadium-substituted Dawson anion ([P_2_W_13_V_5_O_62_]^11−^) and cetrimonium cations [38]. The hybrid was evaluated as a catalyst in the ECODS process of a model fuel using H_2_O_2_ as oxidant. The optimum experimental conditions allowed a sulfur removal of approximately 90% after 45 min and the system could be recycled for seven cycles.

In this work, we have evaluated two quaternary ammonium salts of the monovacant Keggin-type phosphotungstate ([PW_11_O_39_]^7−^) as heterogeneous catalysts in solvent-free ODS processes. The tetrabutylammonium (TBA) and octadecyldimethylammonium (ODA) hybrids, as well as the potassium (K) compound, were tested as catalysts in the desulfurization of a multicomponent model diesel with a total sulfur concentration of 2000 ppm. The proposed ODS system avoids the use of harmful organic solvents by efficiently removing organosulfur compounds under sustainable solvent-free conditions using H_2_O_2_ as oxidant and mixture of 1:1 water/ethanol for the removal of oxidized products. The reusability and stability of the TBA[PW_11_] were investigated for ten consecutive ODS cycles.

## 2. Results and Discussion

### 2.1. Desulfurization of Model Diesel

The ODS studies were performed using a model diesel containing the most refractory sulfur compounds in real diesel (500 ppm or 0.0156 mol dm^−3^ of each): 1-benzothiophene (1-BT), dibenzothiophene (DBT), 4-methyldibenzothiophene (4-MDBT) and 4,6-dimethyldibenzothiophene (4,6-DMDBT) in *n*-octane. The oxidation of these sulfur compounds was performed using H_2_O_2_ as oxidant at 70 °C and catalyzed by the different salts of the monovacant Keggin-type POM (TBA[PW_11_], ODA[PW_11_] and K[PW_11_]), which behaved as heterogeneous catalysts. The catalytic oxidative reaction was carried out under solvent-free conditions, thus avoiding the use of a polar organic solvent. In this system, the oxidation occurs in the diesel phase. Besides, the final extraction of the oxidized products can be conducted with a choice of more sustainable and more cost-effective solvents. This is an important feature for the future industrial application. The organic–inorganic PW_11_ catalysts (TBA[PW_11_] and ODA[PW_11_]) exhibited superior desulfurization performances to the corresponding potassium compound (K[PW_11_]) which showed negligible catalytic activity (Figure 1). This must be due to the affinity of the organic cation of the hybrid catalysts with the model diesel phase, increasing the interphase between model diesel/catalyst/oxidant. On the other hand, the potassium catalyst as inorganic catalyst presents probably an absent interaction with the model diesel. The TBA[PW_11_] hybrid achieved total sulfur conversion at 190 min of reaction while ODA[PW_11_], despite reaching a desulfurization of 83% after only 20 min, was unable to reach total sulfur conversion even after 4 h. These results indicate that the nature of the cation from the catalyst has an important influence in its catalytic performance under a solvent-free system.

The solvent-free systems with the TBA[PW_11_] and ODA[PW_11_] catalysts were submitted to an optimization concerning the amount of oxidant. The same amount of catalyst as used in the ODS solvent-free process optimization (3 μmol) was tested using a H_2_O_2_/S ratio of 3 and 8, at 70 °C. The optimization results obtained for TBA[PW_11_] and ODA[PW_11_] are presented in Figure 2A,B, respectively. It was possible to observe that using the lowest amount of oxidant (H_2_O_2_/S  =  3) the total sulfur conversion could only be achieved with TBA[PW_11_]. For the highest amount of oxidant (H_2_O_2_/S  =  8), total sulfur conversion was attained after 40 min with ODA[PW_11_] while the TBA[PW_11_] system kept a similar kinetic profile, with total sulfur conversion after 190 min of reaction. These results show that the nature of the cation and the length of the carbon chain in active center PW_11_ were revealed to have an important influence in the catalytic performance. The catalyst ODA[PW_11_], which has a longer carbon chain, proved to be more efficient in the sulfur oxidation of the model diesel than TBA[PW_11_], although it used a higher oxidant amount (H_2_O_2_/S = 8). Xu et al. explained this phenomenon through the ability of surfactants with long carbon chains to attract the weakly polar sulfur compounds closer to the surface of POM-hybrids, hence promoting more frequent interactions between sulfur compounds and active species [40]. On the other hand, the higher amount of H_2_O_2_ oxidant needed to achieve complete desulfurization of model diesel catalyzed by ODA[PW_11_] is probably due to the difficult interaction of the polyanion from the catalyst ([PW_11_O_39_]^7−^) with the polar oxidant (aqueous H_2_O_2_), caused by its cationic long carbon chain (ODA^+^). Similar behavior has also been previously reported by our group for biphasic desulfurization systems catalyzed by POM-hybrids and using acetonitrile or [BMIM][PF_6_] as solvents [41,42,43]. Other results show a direct correlation between the carbon chain length of the quaternary ammonium cation in POM-hybrids and their catalytic activity in oxidative desulfurization [34,40,44]. More examples on the application of POM-based hybrids and quaternary ammonium cations in oxidative desulfurization can be found in the literature, with the majority using a biphasic catalytic system. However, more than 2 h are needed to achieve complete desulfurization [1,29,30,31,32,33,34,35,36]. Faster complete desulfurization was reported previously, but using a model diesel containing a single sulfur compound [31,33].

To confirm that PW_11_ is the active center, a new catalyst was prepared with the same cation: the Venturello complex TBA_4_H_3_[PW_4_O_24_] (TBA[PW_4_]). This type of compound has been identified as an intermediate active species in a variety of oxidation reactions formed from the reaction of Keggin-type polyoxometalates with hydrogen peroxide [45,46,47,48,49]. The Venturello TBA[PW_4_] was characterized by FTIR-ATR and ^31^P NMR and the results were in agreement with the previously reported data by Julião et al. (Appendix A) [41,46]. However, under the same experimental catalytic conditions described above, the Venturello complex showed non activity in the solvent-free system, indicating that this is not an active intermediate.

### 2.2. Reusability of the ODS System

The reuse capacity of the TBA[PW_11_] hybrid catalyst was investigated in the solvent-free desulfurization system using 3 μmol of catalyst and H_2_O_2_/S ratio of three for ten consecutive cycles. At the end of each cycle, the desulfurized model diesel was removed and an equal volume of fresh model diesel and oxidant were added to the ODS system, maintaining the same experimental conditions. The results show that the catalyst maintained its performance along ten consecutives ODS cycles reaching complete sulfur conversion after 3 h of reaction (Figure 3). These very promising results surpass the previously reported data for TBA[PW_11_] biphasic systems with acetonitrile or [BMIM][PF_6_] since, in these systems, a decrease in the desulfurization efficiency could be detected during the third consecutive cycle [41].

### 2.3. Catalysts Stability

The robustness of the TBA[PW_11_] hybrid was investigated by recovering the catalyst after catalytic use (ac) and analyzing the solid by FTIR-ATR and ^31^P NMR spectroscopies. The FTIR spectrum of TBA[PW_11_]-ac (Figure 4) still exhibits the characteristic vibrational bands of the starting hybrid. The FT-IR spectrum of TBA[PW_11_] exhibits the bands associated with the ν_as_ (P-O), ν_as_(W-O_t_), ν_as_(W-O_b_-W) and ν_as_(W-O_c_-W) vibrational modes of POM at 1081–1049, 957–952, 892–890 and 814–802 cm^−1^, respectively. The spectrum also displays the characteristic bands associated with the cation, namely the ν_as_(C-H) and ν_s_(C-H) stretches located at 2961, 2936 and 2873 cm^−1^, respectively. In fact, the bands associated with the inorganic POM framework and the organic cations can be observed without major shifts, suggesting that the main structure is preserved [30,41,50]. However, when compared with the spectra of the starting catalyst, the spectrum also shows some additional bands (1280, 1153, 711, 698, 656 and 615 cm^−1^) that are related to the presence of adsorbed sulfur compounds that remain attached to the catalyst, as previously reported by our group [22,23,41].

The recovered solid after catalytic use, TBA[PW_11_]-ac, was also characterized by ^31^P NMR (Figure 5). The spectrum displays a main peak located at δ = −13.86 ppm and another peak at δ = 0.86 ppm. The first peak is shifted when compared with the signal of the starting TBA[PW_11_] (−11.57 ppm) [41] and although the value is close to the ^31^P NMR signal for the plenary Keggin anion ([PW_12_O_40_]^3−^ PW_12_) which is −13.38 ppm, previous catalytic studies showed the absence of catalytic activity from PW_12_ under these same experimental conditions. Nevertheless, it is well documented in the literature that the catalytic activity of the Keggin (PW_12_) anion arises from its decomposition into active peroxotungstate species formed in the presence of excess H_2_O_2_ in biphasic systems [41,51]. In this work, the desulfurization results using TBA[PW_11_] show that the catalyst is able to oxidize the sulfur compounds and additionally without noticeable loss of activity along the reusability experiments. For all of the above, and despite the similar chemical shifts, we believe that the signal in the TBA[PW_11_]-ac spectrum should also correspond to the PW_11_ structure with the lacunary region occupied by peroxo groups (PW_x_O_y_ species). These peroxo-containing species are considered intermediates in the formation of the catalytic active peroxo-compounds, such as the species whose peak appears at δ = 0.86 ppm in the ^31^P NMR spectrum [52].

## 3. Experimental

### 3.1. Materials and Methods

All the reagents used for desulfurization experiments, including 1-benzothiophene (1-BT, Fluka, Buchs, Switzerland), dibenzothiophene (DBT, Aldrich, St. Louis, MO, USA), 4-methyldibenzothiophene (4-MDBT, Aldrich), 4,6-dimethyldibenzothiophene (4,6-DMDBT, Alfa Aesar, Haverhill, MA, USA), *n*-octane (Aldrich), hydrogen peroxide 30% (Aldrich) were purchased from chemical suppliers and used without further purification.

Infrared absorption spectra were recorded for 400–4000 cm^−1^ regions on a Jasco 460 Plus spectrometer (Jasco Analytical Instruments, Easton, PA, USA), using KBr pellets. ^31^P spectra were collected for liquid solutions using a Bruker Avance III 400 spectrometer (Bruker, Freemont, CA, USA) at 161.9 MHz and 298 K. The chemical shifts are given with respect to external 85% H_3_PO_4_. The catalytic reactions were monitored by a Bruker 430-GC-FID gas chromatograph (Bruker, Freemont, CA, USA), using a Supelco capillary column SPB-5 (30 m × 250 µm id.; 25 µm film thickness) and hydrogen as a carried gas (55 cm^3^ s^−1^), using the detector at 290 °C and the injector at 250 °C with an increasing oven temperature from 60 to 270 °C, varying the heating rate between 5 and 20 °C min^−1^.

### 3.2. Synthesis and Characterization of POMs

The syntheses of the potassium, tetrabutylammonium (C_16_H_36_N, abbreviated as TBA) and trimethyloctadecylammonium (C_21_H_46_N, abbreviated as ODA) salts of the lacunary polyoxometalate [PW_11_O_39_]^7−^: TBA_4_H_3_[PW_11_O_39_] (TBA[PW_11_]) and ODA_7_[PW_11_O_39_] (ODA[PW_11_]) and K_7_[PW_11_O_39_]·3H_2_O (K[PW_11_]), respectively, were all previously reported by our research group [41].

### 3.3. Desulfurization of Model Diesel

A model diesel containing the most refractory sulfur compounds in real diesel (500 ppm or 0.0156 mol dm^−3^ of sulfur from 1-BT, DBT, 4-MDBT and 4,6-DMDBT in *n*-octane) was used to investigate the oxidative desulfurization efficiency of TBA[PW_11_], ODA[PW_11_] and K[PW_11_] catalysts. All catalytic experiments were carried out under atmospheric pressure in a closed borosilicate 5 mL reaction vessel loaded with a magnetic stirring bar and immersed in a thermostated oil bath at 70 °C. A solvent-free system was employed by using the model diesel (1 mL), the catalyst (3 µmol) and aqueous H_2_O_2_ 30% (0.24 or 0.61 mmol; H_2_O_2_/S molar ratio = 3 or 8, respectively). Centrifugation was carried out after oxidation to separate the solid catalyst from model diesel. A liquid-liquid extraction using 1:1 ethanol/water ratio was performed when complete oxidation was achieved to remove the oxidized sulfur compounds from the model diesel. The sulfur content in the model diesel was periodically quantified by GC analysis using tetradecane as standard. For the reusability studies, the model diesel was easily removed from the reactor and the catalyst was dried in a desiccator overnight.

## 4. Conclusions

In this work, monovacant polyoxotungstate organic hybrids, TBA[PW_11_] and ODA[PW_11_], were evaluated as catalysts in a solvent-free desulfurization system. The ODA[PW_11_]-based system showed a better performance by reaching a complete sulfur conversion of a multicomponent model diesel (2000 ppm S) after 40 min, although a higher H_2_O_2_/S ratio was needed. In fact, the ODS results revealed that the catalytic activity of the hybrid catalysts is influenced by the type of cation and that long carbon chains favor the desulfurization process using an excess of H_2_O_2_ oxidant. The TBA[PW_11_]-catalyzed system represents a more cost-effective desulfurization option since it was able to remove 96.5% of sulfur compounds in 120 min using a H_2_O_2_/S ratio of only 3. Moreover, this system also showed a remarkable reusability and stability by producing clean diesel after 3 h of reaction for ten consecutive cycles without loss of activity. The proposed mechanism involves the formation of active peroxo-compounds through the interaction of the catalyst with the oxidant (H_2_O_2_). The sustainability and cost-efficiency associated with the proposed solvent-free desulfurization system by avoiding the use of toxic organic solvents and using a reduced amount of oxidant makes it a very promising candidate for application in real diesel.

## Figures and Tables

**Figure 1 molecules-25-04961-f001:**
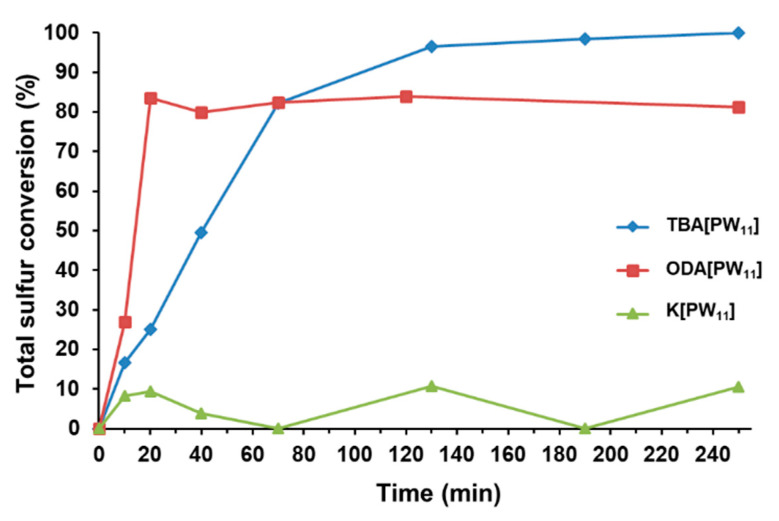
Kinetic profiles for the desulfurization of a model diesel under the solvent-free system, catalyzed by TBA[PW_11_], ODA[PW_11_] or K[PW_11_], using H_2_O_2_ as oxidant (H_2_O_2_/S = 3) at 70 °C.

**Figure 2 molecules-25-04961-f002:**
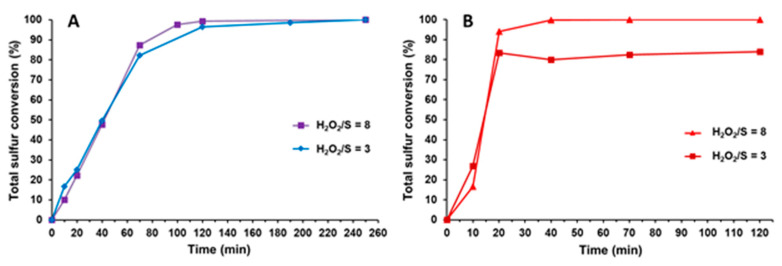
Total conversion data obtained for the oxidation of a multicomponent sulfur model diesel (2000 ppm S) at 70 °C using (**A**) TBA[PW_11_] and (**B**) ODA[PW_11_] catalysts (3 μmol) and different amounts of H_2_O_2_ as oxidant.

**Figure 3 molecules-25-04961-f003:**
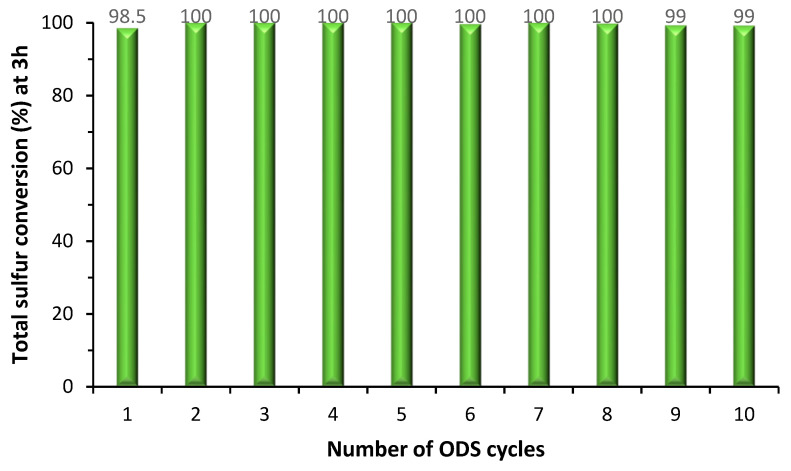
Desulfurization results of a multicomponent model diesel after 3 h for ten consecutive cycles, using a solvent-free system with H_2_O_2_/S  = 3 and catalyzed by TBA[PW_11_].

**Figure 4 molecules-25-04961-f004:**
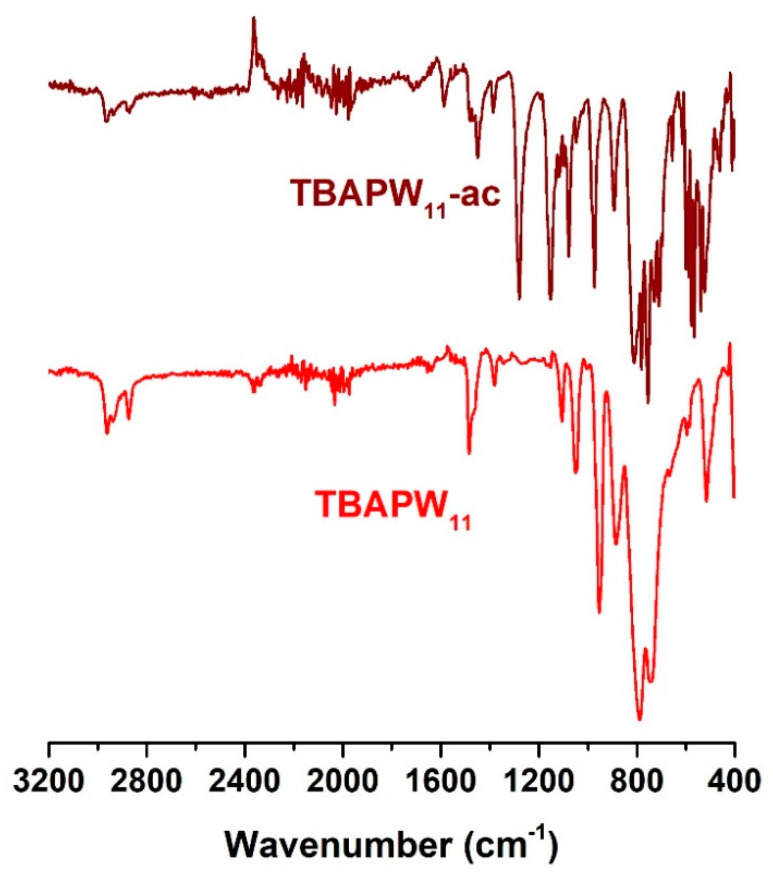
FT-IR spectra of TBAPW_11_ before and after catalytic use (ac).

**Figure 5 molecules-25-04961-f005:**
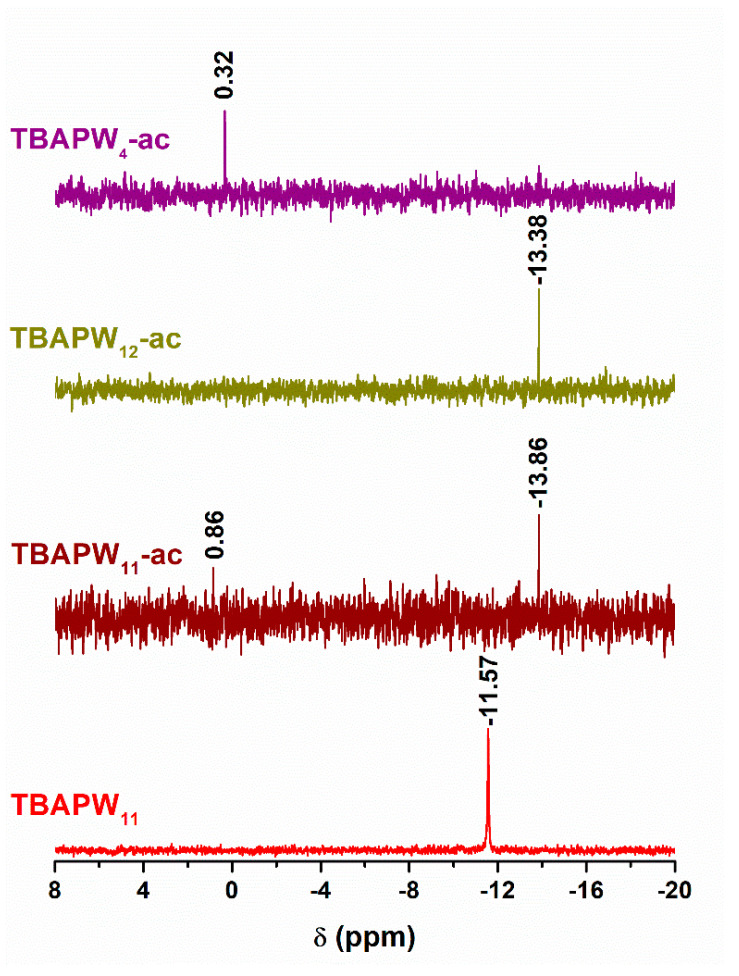
^31^P NMR spectra of as-synthetized TBA[PW_11_] and TBA[PW_11_], TBA[PW_12_] and TBA[PW_4_] after catalytic use (ac).

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
