# Peer review of "Solvent-Free Desulfurization System to Produce Low-Sulfur Diesel Using Hybrid Monovacant Keggin-Type Catalyst"

_molecules, 2020, doi:10.3390/molecules25214961_

Round 1

Reviewer 1 Report

This manuscript deals with the preparation and characterisation of two hybrid quaternary ammonium catalysts based on the monovacant Keggin-type polyoxotungstate and their application in an oxidative, organic solvent-free desulphurisation method to produce diesel oil with a low sulfur content. Since this process (ODS) has high impact on the field of production of sulphur-free fuels, this investigation deserves attention.

In this form, however, this manuscript cannot be published for the following reasons:

1) p. 1 lines 13–42 The authors use many abbreviations (e.g. “ODA”, “TBA”, “ODS”, “DBT”, “4,6-DMBT”), but they are resolved later. You should already express their meaning at the first appearance.

2) p. 2 lines 77–82 In the 2.1. Materials and Methods subsection, the authors should give the exact operational frequencies of the 31P NMR measurements, as well as the temperature profile of the GC analysis.

3) p. 4 lines 129–157 The authors refer to some figures collected in the ESI. Unfortunately, no ESI file is attached to the manuscript (!), so your explanations cannot be followed and understood.

In addition, it would be more informative, if you could insert these figures into the mansuscript.

4) p. 5 line 170 It would be more advantageous, if the conversion values of the ODS reactions were shown in Figure 2.

5) p. 5 line 175 The authors write that “The FTIR spectrum of TBA[PW11]-ac (Figure 3) still exhibits the characteristic vibrational bands of the starting hybrid.”, but no specific values are given and, moreover, there are also no information in the Figure 3. You should improve its lucidity.

6) ps. 7–9  Style of the references does not meet the requirements of journal Molecules. For example, “Aerobic oxidative desulfurization of benzothiophene, dibenzothiophene and 4,6-dimethyldibenzothiophene using an Anderson-type catalyst [(C18H37)2N(CH3)2]5[IMo6O24]. Green Chemistry, 2010. 12(11): p. 1954-1958.” instead of Aerobic oxidative desulfurization of benzothiophene, dibenzothiophene and 4,6-dimethyldibenzothiophene using an Anderson-type catalyst [(C18H37)2N(CH3)2]5[IMo6O24]. Green Chem. 2010, 12, 1954–1958. Please, modify all of them.

7) The English also needs some improvements. There are some typical grammar or typing mistakes:

p. 1 line 14 and elsewhere „ … peroxo-compound …” instead of  peroxo compound

p. 2 line 72 „All the reagents used were used for desulfurization experiments, …” instead of All the reagents used for desulfurization experiments,

      line 74 and elsewhere  „ … n-octane …”  instead of  octane

p. 6 line 189 „… previous catalytic studies have shown the ...” instead of previous catalytic studies showed the

Author Response

are highlighted in blue (relative to reviewer 1). Our responses are as follows:

 Reviewer 1

This manuscript deals with the preparation and characterisation of two hybrid quaternary ammonium catalysts based on the monovacant Keggin-type polyoxotungstate and their application in an oxidative, organic solvent-free desulphurisation method to produce diesel oil with a low sulfur content. Since this process (ODS) has high impact on the field of production of sulphur-free fuels, this investigation deserves attention.

In this form, however, this manuscript cannot be published for the following reasons:

1) p. 1 lines 13–42 The authors use many abbreviations (e.g. “ODA”, “TBA”, “ODS”, “DBT”, “4,6-DMBT”), but they are resolved later. You should already express their meaning at the first appearance.

Response: The authors acknowledge the Referee for this important point, to correct it the authors expressed the meaning of abbreviations at the first appearance.

2) p. 2 lines 77–82 In the 2.1. Materials and Methods subsection, the authors should give the exact operational frequencies of the 31P NMR measurements, as well as the temperature profile of the GC analysis.

Response: The authors acknowledge the Referee for this important observation. This was modified in the manuscript and highlighted in blue.

3) p. 4 lines 129–157 The authors refer to some figures collected in the ESI. Unfortunately, no ESI file is attached to the manuscript (!), so your explanations cannot be followed and understood.

In addition, it would be more informative, if you could insert these figures into the manuscript.

Response: The authors recognised that is more easer to follow and understand the explanations if the figures were insert into the manuscript. As suggested, the authors have inserted a new figure in the manuscript (Figure 2).

4) p. 5 line 170 It would be more advantageous, if the conversion values of the ODS reactions were shown in Figure 2.

Response: The authors acknowledge the Referee for this detail. To better understand the total conversion values of the ODS reactions the authors agreed that it was more advantageous to add them in figure.

5) p. 5 line 175 The authors write that “The FTIR spectrum of TBA[PW11]-ac (Figure 3) still exhibits the characteristic vibrational bands of the starting hybrid.”, but no specific values are given and, moreover, there are also no information in the Figure 3. You should improve its lucidity.

Response: The authors acknowledge the Referee for this observation. In fact, the authors didn´t specify the values of vibrational bands of the starting hybrid in text or in the figure. The authors improved these results by adding the selected FT-IR vibrational bands observed in the spectrum for the TBA[PW11] initial to the manuscript and highlighted in blue.

6) ps. 7–9 Style of the references does not meet the requirements of journal Molecules. For example, “Aerobic oxidative desulfurization of benzothiophene, dibenzothiophene and 4,6-dimethyldibenzothiophene using an Anderson-type catalyst [(C18H37)2N(CH3)2]5[IMo6O24]. Green Chemistry, 2010. 12(11): p. 1954-1958.” instead of Aerobic oxidative desulfurization of benzothiophene, dibenzothiophene and 4,6-dimethyldibenzothiophene using an Anderson-type catalyst [(C18H37)2N(CH3)2]5[IMo6O24]. Green Chem. 2010, 12, 1954–1958. Please, modify all of them.

Response: The authors acknowledge the Referee for this important correction. This was modified in the document according to the requirements of journal Molecules.

7) The English also needs some improvements. There are some typical grammar or typing mistakes:

  1. 1 line 14 and elsewhere „ … peroxo-compound …” instead of  peroxo compound
  2. 2 line 72 „All the reagents used were used for desulfurization experiments, …” instead of All the reagents used for desulfurization experiments,

      line 74 and elsewhere  „ … n-octane …”  instead of  octane

  1. 6 line 189 „… previous catalytic studies have shown the ...” instead of previous catalytic studies showed the

Response: The authors acknowledge the Referee for this important point. The authors corrected the suggested English mistakes. The authors don’t change n-octane because it was the correct name of the solvent used to prepared the model oil.

Reviewer 2 Report

The manuscript deals with the solvent free desulfurisation of fuel, using monovacant Keggin-Type catalyst. The topic of the manuscript is interesting and it currently attracts the interest of the scientific community. The authors are very active in this field. However, although the above premises, this referee has serious concerns about the manuscript and this is the reason why I think that the manuscript should be suitable of publication only after major revision. Some requests are following reported:

1) Page 3, lines 119-122: performing experiments in the presence of the same amount of catalyst and H2O2/S ratio, the AA state that differences observed in the total sulfur conversion can be ascribed to the amount of oxidant. What they mean? The use the same amount of H2O2. In the referee's opinion the above differences can be ascribed to the different nature of the catalyst;

2) Page 4, lines 131-143: The AA state that the ODA[PW11] catalyst is more efficient than TBA[PW11] system as it gives a higher total sulfur conversion in shorter time, but using a higher H2O2/S ratio. Firstly, this referee does not agree with the statement that the use of a higher amount of oxidant can be the evidence of a higher reactivity. Furthermore, in the referee’s opinion is not clear how a catalyst, like ODA[PW11] that is able to solubilise only sulfur compounds, but not the H2O2, can promote the reaction. This section of the manuscript is quite confusing and the authors should better clarify their statements.

3) In the Section 3.3, the authors analyse the catalyst stability. The AA should better indicate IR frequencies corresponding to the POM framework and organic cations to help the reading of less familiar researchers with this topic. Furthermore, they should better explain differences in the 31P chemical shift values of the pristine and reused catalyst, as they are significant.

4) In the Introduction some interesting references about desulfurization of fuel should be mentioned (see for example Green Chem. 2018, 20, 4260; Applied Catal. B Env. 2020, 271, 118936; J. Molecular Liqu. 2020, 309, 113093).

Author Response

 Reviewer 2

The manuscript deals with the solvent free desulfurisation of fuel, using monovacant Keggin-Type catalyst. The topic of the manuscript is interesting and it currently attracts the interest of the scientific community. The authors are very active in this field. However, although the above premises, this referee has serious concerns about the manuscript and this is the reason why I think that the manuscript should be suitable of publication only after major revision. Some requests are following reported:

1) Page 3, lines 119-122: performing experiments in the presence of the same amount of catalyst and H2O2/S ratio, the AA state that differences observed in the total sulfur conversion can be ascribed to the amount of oxidant. What they mean? The use the same amount of H2O2. In the referee's opinion the above differences can be ascribed to the different nature of the catalyst;

Response: The authors acknowledge with the Referee for this important correction. The difference of catalyst activity between TBA[PW11] and ODA[PW11] using 3 umol of catalyst and H2O2/S = 3 is due to the different nature of cation from the catalyst. This was corrected in the manuscript and it is highlighted at brown   

2) Page 4, lines 131-143: The AA state that the ODA[PW11] catalyst is more efficient than TBA[PW11] system as it gives a higher total sulfur conversion in shorter time, but using a higher H2O2/S ratio. Firstly, this referee does not agree with the statement that the use of a higher amount of oxidant can be the evidence of a higher reactivity. Furthermore, in the referee’s opinion is not clear how a catalyst, like ODA[PW11] that is able to solubilise only sulfur compounds, but not the H2O2, can promote the reaction. This section of the manuscript is quite confusing and the authors should better clarify their statements.

Response: The authors would like to clarify the correlation observed between oxidant amount, catalyst nature and their influence in the desulfurization efficiency.

When the authors referred that ODA[PW11] is more efficient that TBA[PW11] because in fact using H2O2/s = 8, faster complete desulfurization of model diesel can be achieved using ODA[PW11] (please check Figures S1 and S2 for H2O2/S = 8, or see the next Figure that conciliates data of Figure S1 and S2). However, this sentence was corrected in the manuscript and it is highlighted at brown.

The authors never referred that ODA[PW11] do not solubilize H2O2. In the manuscript is mentioned a more difficult interaction of the ODA[PW11] catalyst with the aqueous H2O2 oxidant (“difficult interaction of the polyanion from the catalyst ([PW11O39]7−) with the polar oxidant (aqueous H2O2), caused by its cationic long carbon chain (ODA+)”). In fact, an interaction between catalyst and H2O2 oxidant needs to occurs to form the active peroxo species that are probably responsible for sulfur oxidation (see section 3.3 and 31P NMR analysis after catalytic use).

3) In the Section 3.3, the authors analyse the catalyst stability. The AA should better indicate IR frequencies corresponding to the POM framework and organic cations to help the reading of less familiar researchers with this topic. Furthermore, they should better explain differences in the 31P chemical shift values of the pristine and reused catalyst, as they are significant.

Response: The authors acknowledge the Referee for this observation. The attribution of vibrational bands of the hybrid compounds were introduced in the manuscript in page 6 lines 183-187 and it is highlighted at blue. About the 31P NMR analyses after catalytic use, the formation of peroxo-compounds was suggested based in comparison with reported results in the literature [52].

4) In the Introduction some interesting references about desulfurization of fuel should be mentioned (see for example Green Chem. 2018, 20, 4260; Applied Catal. B Env. 2020, 271, 118936; J. Molecular Liqu. 2020, 309, 113093).

Response: The authors acknowledge the Referee for these important references about desulfurization. The authors added that references to the introduction.

Round 2

Reviewer 1 Report

The authors have adapted my suggestions and have corrected the mistakes in their manuscript but, unfortunately, an error still remained in it.

However, the revised manuscript in this form, after correcting the following inaccuracy, can be published.

1) p. 5 line 164 The authors still refer to some figures (S' and S2) collected in the ESI. Unfortunately, no ESI file is attached to the manuscript (!), so your explanations cannot be followed and understood.

Reviewer 2 Report

The authors have raised all referee's concerns. Then, the manuscript is now suitable of publication.